# The Effects of Ocean Acidification and Warming on Growth of a Natural Community of Coastal Phytoplankton

**Bonggil Hyun [1], Ja-Myung Kim [2], Pung-Guk Jang [1], Min-Chul Jang [1], Keun-Hyung Choi [3], Kitack Lee [2], Eun Jin Yang [4], Jae Hoon Noh [5] and Kyoungsoon Shin [1],***

[1] Ballast Water Center, Korea Institute of Ocean Science and Technology, Geoje 53201, Korea; bghyun@kiost.ac.kr (B.H.); pgjang@kiost.ac.kr (P.-G.J.); mcjang@kiost.ac.kr (M.-C.J.)

[2] Division of Environmental Science and Engineering, Pohang University of Science and Technology, Pohang 37673, Korea; jamyung@postech.ac.kr (J.-M.K.); ktl@postech.edu (K.L.)

[3] Department of Ocean Environmental Sciences, Chungnam National University, Daejeon 34134, Korea; keunhchoi@cnu.ac.kr

[4] Division of Polar Ocean Sciences, Korea Polar Research Institute, Incheon 21990, Korea; ejyang@kopri.re.kr

[5] Marine Ecosystem Research Center, Korea Institute of Ocean Science and Technology, Busan 49111, Korea; ijnoh@kiost.ac.kr

*  Correspondence: ksshin@kiost.ac.kr

**Abstract:** An in situ mesocosm experiment was performed to investigate the combined effects of ocean acidification and warming on the coastal phytoplankton standing stock and species composition of a eutrophic coastal area in the temperate-subtropical region. Experimental treatments of natural seawater included three $CO_2$ and two temperature conditions (present control: ~400 μatm $CO_2$ and ambient temperature, acidification conditions: ~900 μatm $CO_2$ and ambient temperature, and greenhouse conditions: ~900 μatm $CO_2$ and ambient temperature +3 °C). We found that increased $CO_2$ concentration benefited the growth of small autotrophic phytoplankton groups: picophytoplankton (PP), autotrophic nanoflagellates (ANF), and small chain-forming diatoms (DT). However, in the greenhouse conditions, ANF and DT abundances were lower compared with those in the acidification conditions. The proliferation of small autotrophic phytoplankton in future oceanic conditions (acidification and greenhouse) also increased the abundance of heterotrophic dinoflagellates (HDF). These responses suggest that a combination of acidification and warming will not only increase the small autotrophic phytoplankton standing stock but, also, lead to a shift in the diatom and dinoflagellate species composition, with potential biogeochemical element cycling feedback and an increased frequency and intensity of harmful algal blooms.

**Keywords:** mesocosm; acidification; warming; picophytoplankton; autotrophic nanoflagellates; diatoms; dinoflagellates

## 1. Introduction

Over the past few centuries, anthropogenic emissions of carbon dioxide ($CO_2$) have resulted in an increase in the atmospheric $CO_2$ concentration from average preindustrial levels of ~280 parts per million volume (ppmv) to more than 400 ppmv in 2014 [1]. Moreover, atmospheric $CO_2$ is predicted to nearly double to 750 ppmv within the next 100 years [2]. The increase in atmospheric $CO_2$ not only leads to ocean warming, via the greenhouse effect, but, also, ocean acidification through an increase in $CO_2$ dissolved in the sea surface, a decrease in sea surface pH, and a decrease in the saturation state of calcium carbonates in the ocean [3]. The Intergovernmental Panel on Climate Change (IPCC)

2014 report projects an additional 1–6 °C temperature increase and ~0.3 pH unit decrease in the sea surface within this century [4]. The current pace of climate change is unprecedented throughout geological history, and some coastal waters will be subjected to temperature increases exceeding 2 °C [5].

These altered oceanographic conditions may impact marine phytoplankton [6,7]. Therefore, numerous laboratory and mesocosm-based experiments have been carried out to examine the effects of ocean acidification on cultured and natural phytoplankton species, and some of the most common observations have involved the fertilization effect of increased $CO_2$ on primary production [8–12]. Recent modeling studies have also suggested that the growth of marine phytoplankton will increase by 40% if $CO_2$ levels increase to 700 ppm [9,13]. These research results showed that, although phytoplankton have evolved carbon-concentrating mechanisms (CCMs) to facilitate the uptake of bicarbonate ($HCO_3^{-1}$) and its conversion to $CO_2$, increased $CO_2$ concentrations may still be beneficial for their growth, because they could help to reduce the metabolic cost of CCMs [3]. However, since CCM efficiencies differ between phytoplankton species, subtle changes in or neutral effects on the phytoplankton standing stock and primary production have also been reported [14–16].

In addition to ocean acidification, ocean warming is probably the most widely recognized consequence of climate change. Previous studies have shown that ocean warming can both enhance phytoplankton primary production to a greater extent than a $CO_2$ concentration increase [17,18] and affect the phytoplankton size distribution, which triggers a dominance of nanophytoplankton and picophytoplankton (PP) [19,20]. Ocean warming also strongly influenced the phytoplankton community structure and species composition by altering biophysical activities across all trophic levels [17,18]. In the future, ocean acidification and warming are expected to occur simultaneously, and thus, it is predicted that there will be changes in the phytoplankton community structure and species composition compared with the current marine ecosystem. This is assumed to have implications for the turnover of organic matter and biogeochemical cycling and might influence the efficiency of the biological pump [20]. Ideally, their effects on natural phytoplankton communities should be investigated in unison, as the combined effects of ocean acidification and warming might be completely different from that of either stressor alone [17,21].

Mesocosm experiments, comprising natural plankton communities, are ideal platforms for assessing the potential effects of increased $pCO_2$ and temperature, as they allow for species interaction and competition in a quasi-natural environment [22,23]. To do this, we conducted a mesocosm experiment on the southern coast of Korea under control (i.e., present), ocean acidification (i.e., high $CO_2$), and combined ocean acidification and warming (i.e., greenhouse) conditions to investigate the responses of natural phytoplankton communities from community (picophytoploankton (PP), autotrophic nanoflagellate (ANF), diatoms (DT), and dinoflagellates (DINO)) to species level (DT and DINO). Our research results are then discussed in light of the following questions: (i) What are the effects of high $pCO_2$ on the growth of phytoplankton in eutrophic coastal waters? (ii) How do phytoplankton standing stock and species composition change under the greenhouse treatments? and (iii) Is the growth of small-sized phytoplankton and heterotrophic dinoflagellates promoted under acidification and greenhouse treatments? The increases in water temperature and $pCO_2$ applied during our mesocosm experiments were within the range projected by the IPCC 2007 for the end of this century [24].

## 2. Materials and Methods

### 2.1. Experimental Design

The study was conducted for 21 days (from 21 November to 11 December 2008) at the South Sea Institute of the Korea Institute of Ocean Science and Technology in Jangmok (34.6° N, 128.5° E) near the southern coast of Korea. The experiment involved mesocosm enclosures of 2400 L (1 m in diameter and 3 m in height), in triplicate, which were used to simulate three sets of conditions based

on model projections under the A2 Scenario of the Intergovernmental Panel on Climate Change Special Report on Emissions Scenarios [25]: (i) present control (~400 μatm $CO_2$ and ambient temperature), (ii) acidification conditions (~900 μatm $CO_2$ and ambient temperature), and (iii) greenhouse conditions (~900 μatm $CO_2$ and ~3 °C warmer than ambient temperature) (Figure 1). The target seawater $pCO_2$ levels and temperature increase were achieved by mixing $CO_2$-saturated seawater with ambient seawater and by circulating warm water through tubing wrapped around the seawater mixers in the enclosures, respectively. A detailed description can be found elsewhere [26,27]. Prior to filling the enclosures, 13.5 tons of seawater was passed through a mesh with a pore size of 100 μm to remove large heterotrophic grazers. To initiate the development of a phytoplankton bloom, the same amounts of nutrients were added to each enclosure on day 0, yielding initial concentrations of ~50 μmol $L^{-1}$ Si ($Si(OH)_4$), ~2.5 μmol $L^{-1}$ P ($HPO_4^{2-}$), and ~33 μmol $L^{-1}$ N ($NO_3^- + NO_2^-$). To enhance the distribution homogeneity of the phytoplankton populations, we gently mixed the seawater for 20 min prior to daily sampling using bubble-mediated mixers [27]. All the enclosures were sampled daily at 13:00 h from a ~1-m depth using a fluid metering pump to avoid rupturing cells.

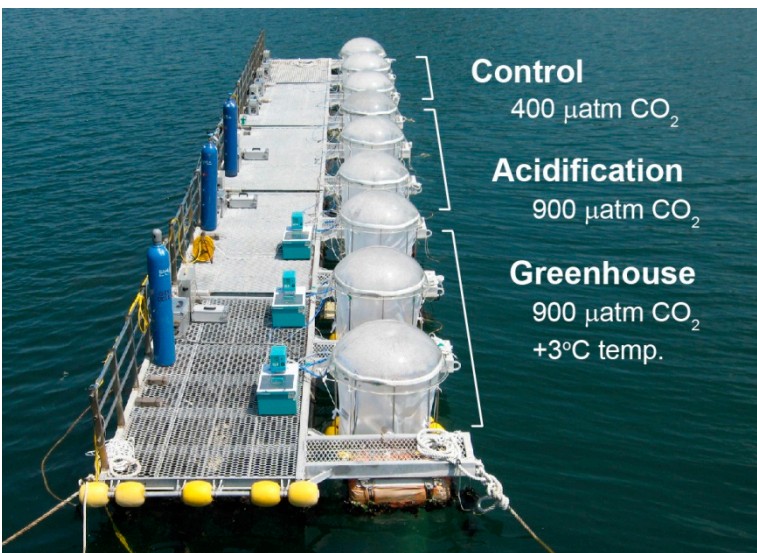

**Figure 1.** Photograph of the floating raft, $pCO_2$ concentration regulation units, temperature regulation units, and enclosures of the mesocosm facility. Design and performance of the facility have been described in detail elsewhere [26,27].

### 2.2. Measurement of Parameters

Temperature and salinity were measured using a multiparameter water quality sonde (YSI-6600). Samples for nutrient (Si, P, and N) analyses were filtered through a 25-mm syringe filter (0.45-μm pore size) into 50-mL acid-cleaned polyethylene tubes and stored at −80 °C until analysis. Nutrient concentrations were analyzed using an autoanalyzer (Quickchem 8000 Flow Injection Analyzer; Lachat, Loveland, CO, USA) following the methods of Parsons et al. [28]. Seawater $pCO_2$ levels were calculated using the total alkalinity and total dissolved inorganic carbon ($[CO_2] + [HCO_3^-] + [CO_3^{2-}]$) concentrations, which were respectively measured using potentiometric and coulometric titration methods within the VINDTA system (MARIANDA, Kiel, Germany), using the carbonic acid dissociation constants that were provide to be consistent with laboratory and field measurements [29–34].

To identify the dominant phytoplankton groups and determine the cell abundances, an aliquot (~500 mL) of seawater sample was immediately preserved with Lugol's solution (5% final concentration), and the phytoplankton cells were concentrated in sedimentation chambers for ≥48 h. The composition of phytoplankton species was assessed using light microscopy (Zeiss, Axioplan 2) at a magnification of ×200–1000. Species identification was based on Rines [35] and Tomas [36]. The DT and DINO

were counted using a light microscope equipped with a Sedgewick Rafter chamber. To estimate the abundance of autotrophic DINO (ADF) and heterotrophic DINO (HDF), 1 L of seawater sample was preserved with formalin (2% final concentration), and the fixed cells were concentrated at 4 °C in the dark for ≥48 h. An aliquot (2–5 mL) of the concentrated sample was further sedimented in a Sedgewick-Rafter chamber and stained with 4′,6-diamidino-2-phenylindole (DAPI; 5% final concentration). The ADF and HDF were enumerated with an inverted epifluorescence microscope (Olympus IX 70) and distinguished by UV and blue-light excitation, which detected the autofluorescence of chlorophyll pigments. Autotrophic nanoflagellates (ANF) were preserved by the addition of glutaraldehyde (1% final concentration). An aliquot (10–30 mL) of the fixed sample was filtered through a black nucleopore filter (0.45-μm pore size); the cells on the filter were stained with DAPI (5 mg.mL$^{-1}$ final concentration) and proflavin (0.33%) and counted using epifluorescence microscopy (Nikon type 104) at a magnification of ×600–1000. To determine the abundance of PP, ~10 mL of the sample was preserved by the addition of paraformaldehyde (1% *w/v*) and glutaraldehyde (0.05% *v/v*) and stored frozen at −80 °C until analysis. The frozen samples were thawed in batches immediately prior to analysis. The cells were concentrated with a black polycarbonate filter (0.2-μm pore size), stained with primulin, and enumerated using epifluorescence microscopy. The growth of the phytoplankton was checked by measuring fluorescence unit (FSU) (Turner Designs 10-AU) daily after incubation, and the growth rates, in doubling per day (μ), were calculated by applying the following equation:

$$\mu = (\log_2 N_t - \log_2 N_0)/t \qquad (1)$$

where t is the length of incubation (days), $N_0$ is the cell abundance at the start of the exponential phase, and $N_t$ is the cell abundance at the end of the exponential phase.

### 2.3. Statistical Analysis

Analysis of variance (ANOVA) was used to determine whether changes in the abundances of phytoplankton groups and species in response to changes in seawater $pCO_2$ and temperature were statistically significant. All datasets met the assumptions of normality and homogeneity of variance. If the ANOVA test identified a significant difference between conditions (*p*-value < 0.05), Scheffe's post-hoc test was applied (SPSS Inc., Chicago, IL, USA).

## 3. Results

### 3.1. Carbonate Parameters and Temperature Dynamics

The seawater $pCO_2$ values for the acidification and greenhouse treatments (target ~900 μatm) and the control (~400 μatm) remained higher than the respective initial levels of 860 and 370 μatm over eight (for acidification) or 12 (for the other two conditions) days (Figure 2a). Due to the influence of heterotrophic activity, which released $CO_2$ during the pre-bloom period (days 0–5), the seawater $pCO_2$ increased by approximately 30–47% of the initial values. As autotrophic production dominated during the bloom period (days 6–14), $pCO_2$ significantly decreased in all nine enclosures ($\Delta pCO_2$ 270, 610, and 387 μatm for the control, acidification, and greenhouse treatments, respectively), and the values remained approximately constant thereafter. Since the decrease in seawater $pCO_2$ associated with biological production was compensated for only by the influx of $CO_2$ from the atmospheres of the enclosures (treated with the same as the target levels of $pCO_2$), the changes in seawater $pCO_2$ did not rapidly recover. However, large differences in $pCO_2$ between the control and other treatments (approximately 220 and 570 μatm for the acidification and greenhouse treatments after the bloom, respectively) were maintained until the end of the experiment. There was a higher $pCO_2$ level in the greenhouse condition than the acidification condition, which was due to the temperature-induced change in seawater $pCO_2$ (4% $pCO_2$ increase °C$^{-1}$) [37,38] and biological activities (see below).

During the experiment, the pH values were 7.93–8.41 in the control treatments, 7.56–8.06 in the acidification treatments, and 7.59–7.83 in the greenhouse treatments, showing a trend opposite to

the seawater $pCO_2$ values (Figure 2b). The ambient seawater temperature in the control gradually decreased from 14 °C on day 0 to 12 °C on day 20, with decreasing air temperature from fall to early winter (Figure 2c). In the greenhouse treatment, an ~2.5 °C elevation in seawater temperature was achieved and maintained throughout the experiment.

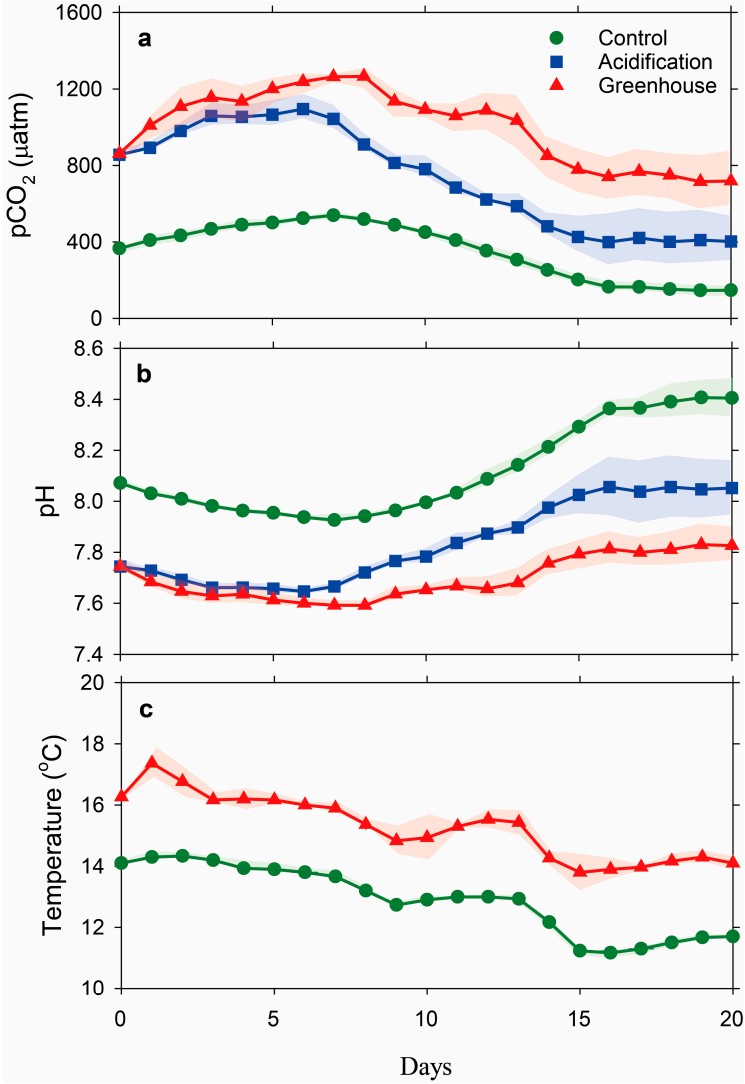

**Figure 2.** Changes in (**a**) seawater $pCO_2$; (**b**) pH; and (**c**) seawater temperature in the control, acidification, and greenhouse enclosures during the study period. The seawater temperatures in the control and acidification treatments are approximately the same, overlapping with a single graph (control). The green, blue, and red symbols represent the control, acidification, and greenhouse conditions, respectively. The color shading represents the standard deviation (1 σ) from the mean for the replicate mesocosms.

### 3.2. Bloom Development

Following the addition of Si, N, and P to the mesocosm enclosures on day 0, the fluorescence (an indicator of autotrophic phytoplankton biomass) slowly increased during the pre-bloom period (days 0–5), showing no differences among the control and treatment enclosures (Figure 3a).

During the bloom period (days 6–14), the biomass of phytoplankton started to exponentially increase. The upward trend in fluorescence was related to the downward trend in the concentrations of added nutrients (Figure 3b–d). In all enclosures, the concentrations of N and P decreased with a mean daily utilization ratio of 17.0 ($\Delta$N/$\Delta$P) until they reached below the level of limitation (N < 1 $\mu$mol L$^{-1}$ and $p$ < 0.2 $\mu$mol L$^{-1}$) on day 14 (in the acidification and greenhouse

treatments) and 15 (in the control) (Figure 3b,c) (ANOVA, P: $p < 0.05$ and N: $p = 0.072$). The rates of decrease in the N and P concentrations were higher under the high-$pCO_2$ conditions (i.e., the acidification and greenhouse treatments) than the control conditions during the bloom period. The peak fluorescence (~12 FSU) was found in the acidification treatment on day 12 and was 1.7 times higher than that in the greenhouse treatment, indicating that the high temperature did not have a synergistic effect (Figure 3a).

During the post-bloom period (days 15–20; N and P not detectable), the exhaustion of N and P was followed by a continuous Si uptake until the end of the experiment (5.98 ± 4.38 μmol L$^{-1}$ Si on day 20) at a slower rate than previously seen, except in the control (Figure 3d) (ANOVA, $p < 0.001$). The largest decrease in the concentration of Si in the control coincided with the highest production of biomass in the same conditions during the post-bloom period.

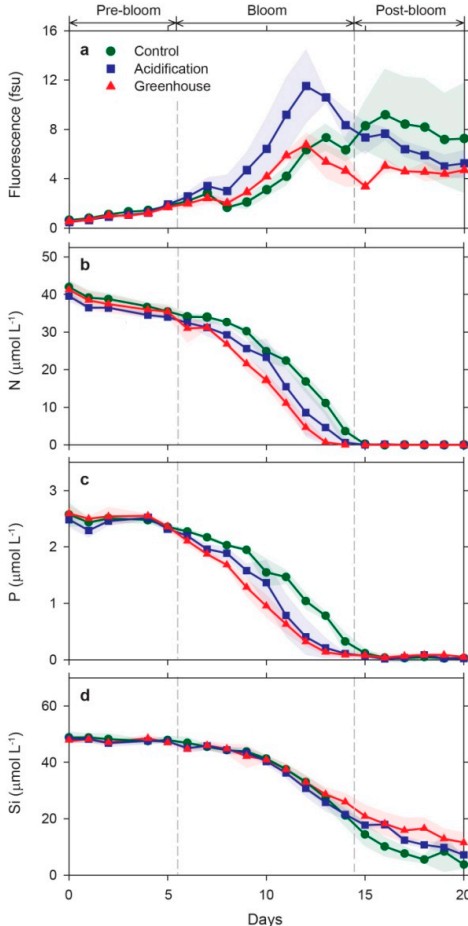

**Figure 3.** Changes in (**a**) fluorescence; (**b**) N; (**c**) P; and (**d**) Si concentrations during the study period. The experiment period was divided into the pre-bloom (days 0–5), the bloom (days 6–14), and the post-bloom periods (days 15–20). The symbols and the color shading represent the same as shown in Figure 2.

*3.3. Population Dynamics of Major Phytoplankton Classes*

During the bloom period, the contribution of PP, ANF, DT, and DINO to the total phytoplankton communities in all enclosures was 70%, 20%, 10%, and 0.2%, respectively (Figure 4). The blooms of PP, ANF, and DT under the high $pCO_2$ conditions (the acidification and greenhouse treatments) were initiated a few days earlier than those in the control, and their abundances increased at faster rates (Figure 4 and Table 1), whereas the abundance of ADF showed no substantial increase in any enclosure during the same period. As inferred from the total biomass in the greenhouse conditions

(Figure 3a), the elevated temperature with high $pCO_2$ did not synergistically enhance the growth of the major phytoplankton classes (Figure 4); rather, the high temperature of the greenhouse treatment lowered both the growth rate (Table 1) and the dominance of ANF and DT compared with the normal temperature of the acidification treatment, and only PP growth benefited from the higher temperature during the bloom period.

During the post-bloom period, the net cell abundances of PP, ANF, and DT decreased, and they maintained higher concentrations in the acidification than the other conditions (Figure 4). In contrast, the abundance of DINO, specifically HDF, which responded from day 11 when the bloom of other phytoplankton classes reached a plateau, continuously increased in all enclosures until the end of the experiment, reaching a 2.5-times higher cell abundance under the high-$CO_2$ conditions than the control (Figure 4j–l).

Overall, a statistically significant successional shift in major phytoplankton classes was found to be induced by changes in the $pCO_2$ levels and temperature (Table 1). During the bloom period, PP dominated the community in all treatment conditions over 60%, and their mean cell abundances increased in the order of the greenhouse, acidification, and control conditions (Figure 4a–c). The relative abundances of ANF and DT were higher in the acidification conditions (ANF: 20%; $p < 0.01$ and DT: 18%; $p < 0.001$) relative to those in both the control (ANF: 18% and DT: 8%) and greenhouse conditions (ANF: 14% and DT: 7%) (Figure 4d–i). During the post-bloom period, the relative abundance of DINO was higher in the greenhouse conditions (8%) than the other two conditions (<3%) (Figure 4j–l; $p < 0.001$).

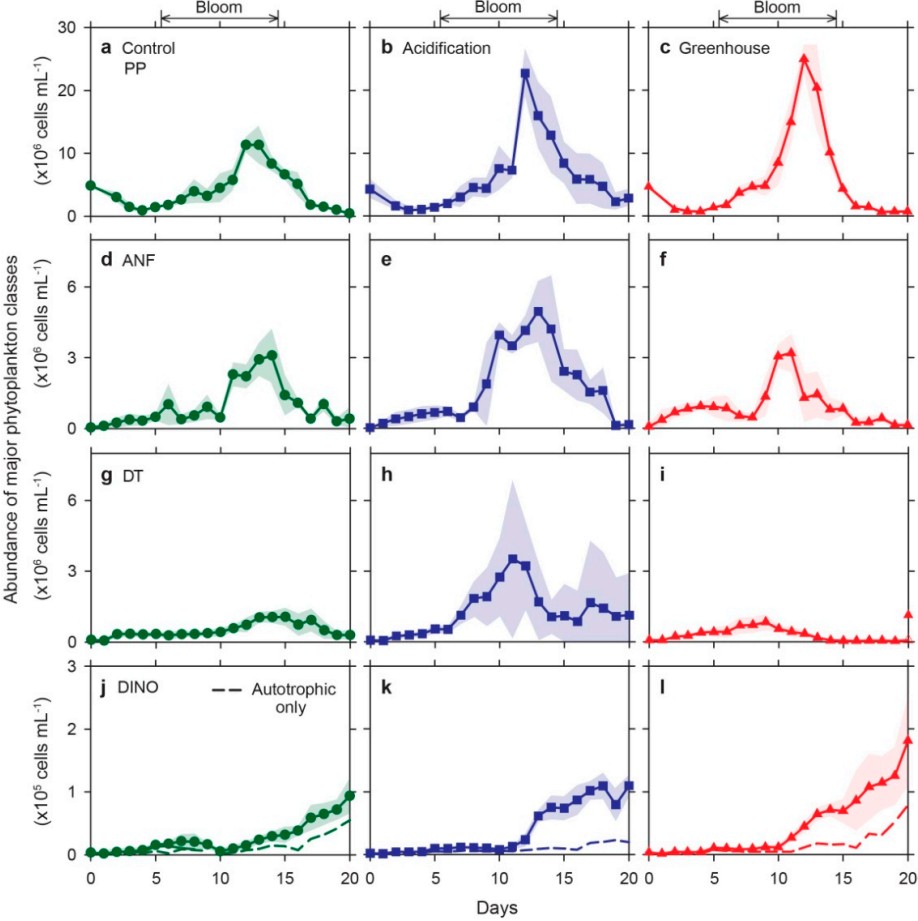

**Figure 4.** Changes in the cell abundance of major phytoplankton classes during the study period: (**a–c**) picoplankton; (**d–f**) autotrophic nanoflagellates; (**g–i**) diatoms; and (**j–l**) dinoflagellates. The symbols and the color shading represent the same as shown in Figure 2.

**Table 1.** Effects of $pCO_2$ and temperature manipulation on specific growth rates and the mean cell abundance of phytoplankton groups and dominant species in the control, acidification, and greenhouse enclosures. The mean cell abundance and its statistical results were calculated during the bloom period (for PP, ANF, and DT) and the post-bloom period (for DINO).

| Phytoplankton Classes | Species | Specific Growth Rate ($d^{-1}$) ± s.d. | | | Mean Cell Abundance ($10^3$ Cells $L^{-1}$) ± s.d. | | | |
|---|---|---|---|---|---|---|---|---|
| | | Control | Acidification | Greenhouse | Control | Acidification | Greenhouse | F Value |
| Picophytoplankton (PP) | | 0.26 ± 0.02 | 0.40 ± 0.02 | 0.42 ± 0.02 | 5864 ± 1209 [A] | 8894 ± 1332 [A,B] | 10,421 ± 1,595 [B] | 3.3 * |
| Autotrophic nanoflagellate (ANF) | | 0.22 ± 0.05 | 0.26 ± 0.03 | 0.26 ± 0.13 | 1541 ± 261 [A] | 2743 ± 392 [B] | 1435 ± 159 [A] | 6.8 ** |
| Diatoms (DT) | | 0.26 ± 0.01 | 0.32 ± 0.15 | 0.26 ± 0.13 | 576 ± 141 [A] | 1976 ± 1297 [B] | 478 ± 115 [A] | 19.3 *** |
| | *Skeletonema* spp. | 0.60 ± 0.25 | 0.73 ± 0.11 | 0.62 ± 0.04 | 127 ± 53 [A] | 1298 ± 673 [B] | 315 ± 111 [A] | 21.7 *** |
| | *Chaetoceros socialis* | 0.48 ± 0.12 | 0.65 ± 0.14 | 0.32 ± 0.23 | 59 ± 32 [A] | 211 ± 311 [B] | 5 ± 4 [A] | 6.3 ** |
| | *Cerataulina* spp. | 0.49 ± 0.12 | 0.41 ± 0.00 | 0.31 ± 0.06 | 66 ± 12 [B] | 17 ± 17 [A] | 5 ± 1 [A] | 14.3 *** |
| | *Chaetoceros decipiens* | 0.52 ± 0.04 | 0.39 ± 0.08 | 0.44 ± 0.04 | 74 ± 25 [B] | 27 ± 12 [A] | 10 ± 3 [A] | 17.6 *** |
| | *Eucampia zodiacus* | 0.70 ± 0.25 | 0.53 ± 0.03 | 0.46 ± 0.06 | 24 ± 7 [B] | 9 ± 3 [A] | 2 ± 0 [A] | 7.3 ** |
| Dinoflagellates (DINO) | | 0.20 ± 0.03 | 0.35 ± 0.02 | 0.27 ± 0.02 | 60 ± 23 [A] | 93 ± 16 [B] | 114 ± 39 [B] | 14.1 *** |
| | *Gyrodinium* spp. | 0.41 ± 0.06 | 0.50 ± 0.14 | 0.41 ± 0.15 | 16.8 ± 0.6 [A] | 45.8 ± 0.8 [C] | 26.9 ± 0.4 [B] | 40.7 *** |
| | *Protoperidinium bipes* | 0.50 ± 0.06 | 0.58 ± 0.10 | 0.62 ± 0.23 | 4.6 ± 0.4 [A] | 10.9 ± 0.4 [A] | 26.7 ± 0.4 [B] | 7.7 * |
| | *Nematodinium armatum* | 0.59 ± 0.15 | 0.56 ± 0.27 | 0.70 ± 0.06 | 7.9 ± 0.2 [A,B] | 5.6 ± 0.1 [A] | 16.3 ± 0.3 [B] | 5.1 * |
| | *Prorocentrum dentatum* | 0.51 ± 0.21 | 0.63 ± 0.15 | 0.71 ± 0.08 | 2.8 ± 0.0 | 2.9 ± 0.1 | 4.7 ± 0.1 | n.s. |

s.d. standard deviations. Statistical results were analyzed for the mean cell abundances using one-way ANOVA and Scheffe's post-hoc test: F-values ($p$) indicates the level of significance (n.s. not significant, * $p < 0.05$, ** $p < 0.01$, and *** $p < 0.001$). Superscripted letters ([A,B] and [C]) indicate significant differences between conditions.

*3.4. Species-Specific Growth Response of Diatoms and Dinoflagellates*

The population of DT, one of the dominant phytoplankton classes for which the species-specific responses were tested, consisted of a group of small chain-forming species (e.g., *Skeletonema* spp. and *Chaetoceros socialis*) and other large diatom species (*Cerataulina* spp., *Chaetoceros decipiens*, and *Eucampia zodiacus*). *Skeletonema* spp. (~246 $\mu m^3$), which accounted for up to 75% of the total diatom abundance, grew seven times faster in the acidification treatment than in the control conditions during the bloom period (Figure 5a and Table 1). Similarly, *Chaetoceros socialis* (77 $\mu m^3$) exhibited a significantly positive-growth response to the high $pCO_2$ levels in the same treatment (Figure 5b and Table 1). By contrast, all the other large, dominant diatom species (1725–30,906 $\mu m^3$, with approximately one order of magnitude lower mean cell abundances than the small diatom species) showed their highest cell abundances and growth rates in the control during the late-bloom period (Figure 5c–e and Table 1). In general, the higher temperature in the greenhouse treatment negatively affected the growth of all diatom species (regardless of their size) during the whole experimental period, except *Skeletonema* spp., which revealed slightly enhanced growth during the bloom period compared with the control (Figure 5a).

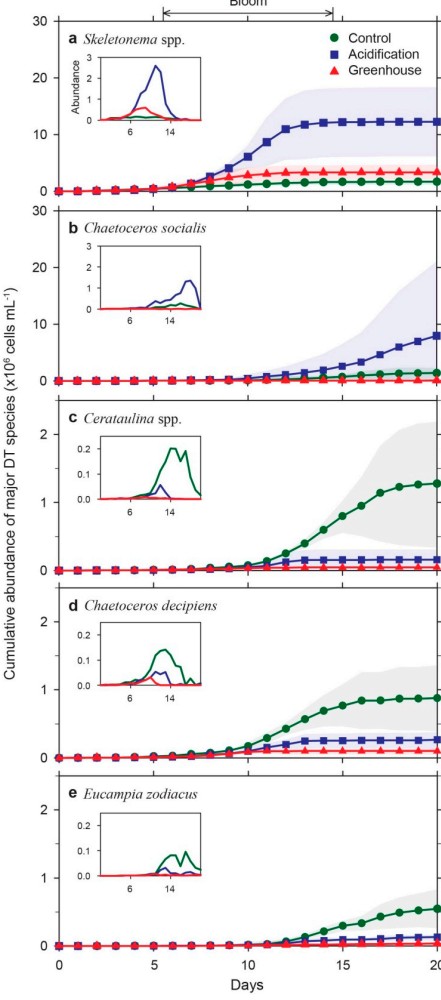

**Figure 5.** Changes in the cumulative cell abundance of major diatom species during the study period: (**a**) *Skeletonema* spp.; (**b**) *Chaetoceros socialis*; (**c**) *Cerataulina* spp.; (**d**) *Chaetoceros decipiens*; and (**e**) *Eucampia zodiacus*. (Insets) Changes in the cell abundance during the same period. The symbols and the color shading represent the same as shown in Figure 2.

The abundance of HDF, accounting for ~50–80% of the total DINO population (Figure 4j–l), increased at considerably higher rates in the acidification and greenhouse treatments compared with the control conditions as a result of the increased abundance of potential prey in the high-$pCO_2$ conditions (Figure 6a). The major DINO species—*Gyrodinium* spp., *Protoperidinium bipes*, *Nematodinium armatum*, and *Prorocentrum dentatum*—showed enhanced growth in the acidification and greenhouse treatments from the late-bloom to the post-bloom periods (Figure 6b–e). In the greenhouse conditions, the higher temperature with $pCO_2$ stimulated the growth of the majority of the dominant DINO species (Figure 6c–e).

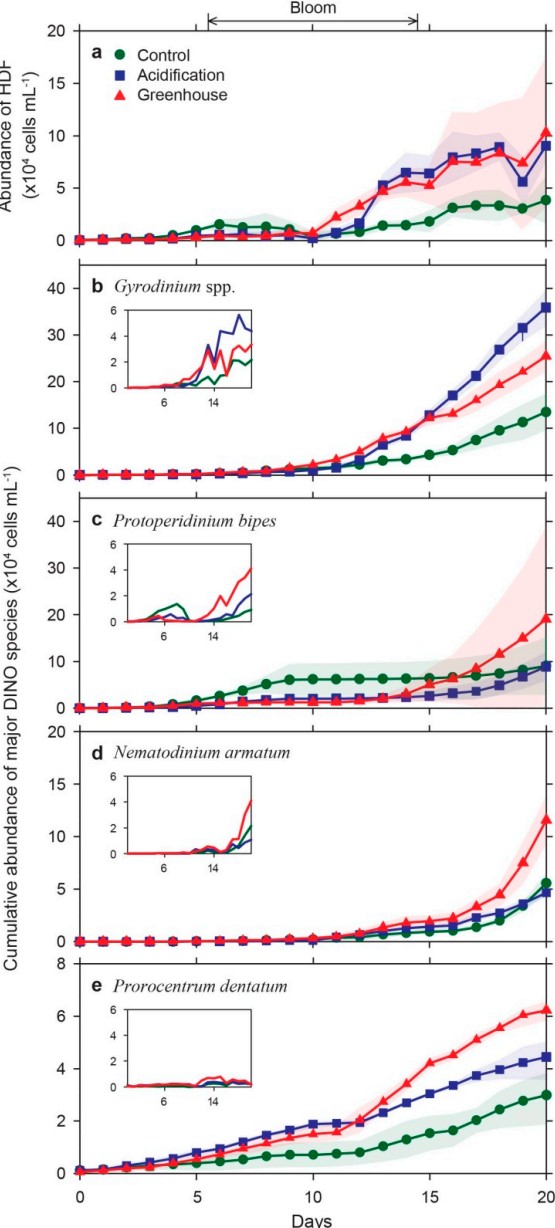

**Figure 6.** Changes in (**a**) the abundance of HDF and (**b–e**) the cumulative cell abundance of major dinoflagellate species during the study period: (**b**) *Gyrodinium* spp.; (**c**) *Protoperidinium bipes*; (**d**) *Nematodinium armatum*; and (**e**) *Prorocentrum dentatum*. (Insets) Changes in the abundance during the same period. The symbols and the color shading represent the same as shown in Figure 2.

## 4. Discussion

The major finding of the present study is that the growth of autotrophic phytoplankton populations (ANF, PP, and small chain-forming DT species) may benefit under future oceanic conditions.

$CO_2$ could be a limiting resource for most phytoplankton communities, as the $CO_2$ concentration is usually below that required for the half-saturation of Ribulose-1, 5-bisphosphate Carboxylase Oxygenase (RuBisCO), the core carbon-fixing enzyme in photosynthesis [3,39]. To overcome the $CO_2$ limitation, therefore, most phytoplankton utilize active inorganic carbon-acquisition mechanisms for photosynthesis (carbon-concentrating mechanisms) based on the active uptake of $CO_2$ and/or $HCO_3^-$ from the environment [40,41]. This hypothesis is consistent with our phytoplankton community response results: shifts in the $CO_2$-dependent phytoplankton communities occurred, but an increased $CO_2$ concentration had a positive effect on their growth. In particular, autotrophic phytoplankton populations, including PP, ANF, and DT, were more abundant in the simulated ocean acidification conditions compared with the control conditions (Table 1 and Figure 4). Until recently, whether phytoplankton will benefit from the projected increase in $CO_2$ was controversial, as $CO_2$ responses vary between phytoplankton groups and species [21,41]. However, there have been clear cases of positive responses by phytoplankton, from small picoplankton to large DT [11,12,42,43], and previous studies have shown that increased $CO_2$ concentrations in coastal eutrophic water enhance the production and productivity of micro-, nano-, pico-sized phytoplankton [44]. Therefore, our results from the eutrophic coastal waters of temperate-subtropical climate regions support the general proposition that increasing $CO_2$ can enhance the growth of the autotrophic phytoplankton community.

During the bloom period in the acidification conditions, the growth responses of the DT species among the autotrophic phytoplankton groups showed that the small chain-forming DT species grew well, whereas large DT species, including *Cerataulina* spp., *Chaetoceros decipiens*, and *E. zodiacus*, did not. These results contrast with previous findings that large phytoplankton species outcompete smaller ones in high-$CO_2$ ocean environments with eutrophication [44]. Therefore, we questioned whether the high concentrations of $CO_2$ in the acidification treatment were causing the lower growth rates of large DT species, and we hypothesized that this growth reduction might be due to nutrient availability, as, along with the $CO_2$ concentration, this is one of the most important parameters for phytoplankton growth. The concentrations of N and P decreased rapidly in the acidification conditions during the bloom period compared with in the control conditions: the small chain-forming DT *Skeletonema* spp., with high N and P uptakes, contributed to more than 50% of the total DT abundance on days 9 and 12. This led us to believe that *Skeletonema* spp. pre-emptively absorbed large amounts of nutrients and $CO_2$ for growth; therefore, the growth of other species, such as large DT species, was inhibited as a result of nutrient-depletion in the acidification conditions. Although additional mesocosm experiments are required to verify the direct effect of $CO_2$ on the growth of large DT species, our results suggest that these would flourish as well in acidification conditions as in the control conditions if resources essential for growth were not limited.

Following the bloom periods, shifts in the phytoplankton community were significantly influenced by the abundance of HDF (Figure 6a). We excluded mesozooplankton from the study, because they can significantly affect the growth of the phytoplankton community, and thus, HDF were the main microzooplankton grazers. Previous research suggested that small autotrophic phytoplankton are generally considered prey sources for microzooplankton, and microzooplankton grazing rates are often well-correlated with standing stocks of algal prey [45]. Increases in HDF abundance in both the acidification and greenhouse conditions may also be related to the increased prey availability (estimated based on the abundance of PP, ANF, and small DT species) (Table 1 and Figure 4). Moreover, the increased HDF metabolic rates in the greenhouse conditions may allow HDF to take advantage of an increased prey encounter rate [46]. Therefore, top-down processes by HDF may control both the intensity and duration of small autotrophic phytoplankton blooms in future coastal environments.

Changes in species compositions within the DT and DINO groups were observed between the treatments. Of the 77 species observed during the whole experimental period, 13 dominant

phytoplankton (five DT and four DINO) species were selected and analyzed for their response against the rising $CO_2$ and temperature (Table 1). Except for *Prorocentrum dentatum* ($p$ = 0.05), there were statistically significant differences between the treatments (Table 1). First, *Skeletonema* spp. and *Chaetoceros socialis* showed positive responses in the acidification conditions, but they showed no or poor growth in the greenhouse conditions compared with in the control conditions (Figure 5a,b). The lower abundance of *Chaetoceros socialis* in the greenhouse conditions is understandable, because this species is known to prefer cold-water conditions [47]; however, the low growth of *Skeletonema* spp. was not fully understood and was only inferred through the grazing activities of HDFs, as the highest growth rates for *Skeletonema* spp. were found at temperatures between 20 °C and 30 °C [48]. In contrast, the large, centric DT species *Cerataulina* spp., *Chaetoceros decipiens*, and *E. zodiacus* did not exhibit positive growths in the acidification and greenhouse condition experiments (Figure 5c–e). The most dominant dinoflagellate species (*Gyrodinium* spp., *Protoperidinium bipes*, *N. armatum*, and *Prorocentrum dentatum*) exhibited positive growth in all the future oceanic conditions (Table 1 and Figure 6). From this study, we found that, as well as the $CO_2$ and water temperature, nutrient concentrations affected DT growth, and prey abundance affected DINO growth. Therefore, predicting the response of DT and DINO communities in eutrophic coastal waters to future climate scenarios will depend on the physio-ecological characteristics of each species, although, overall, it appears that small DT and HDF species will flourish.

Our results showed that increased $CO_2$ concentrations will benefit the growth of small autotrophic phytoplankton communities. We also found that higher water temperatures had a negative effect on the growth of most autotrophic phytoplankton groups/species, except for PP, compared with the acidification conditions. The proliferation of small phytoplankton groups also affected the abundance of HDF; thus, HDF were abundant in the future oceanic conditions.

**Author Contributions:** B.H.: sampling, diatom, and dinoflagellate analysis and writing—original draft. J.-M.K. and K.L.: conceptualization and writing—review and editing. P.-G.J., M.-C.J., and K.-H.C.: chlorophyll and nutrient analysis and writing—review. E.J.Y.: autotrophic nanoflagellate and heterotrophic dinoflagellate analysis. J.H.N.: picophytoplankton analysis. K.S.: supervision, conceptualization, and funding acquisition. All authors have read and agreed to the published version of the manuscript.

**Funding:** This research was supported by the research program of the Korea Institute of Ocean Science and Technology (PE99812).

**Acknowledgments:** The authors would like to thank Woo-Jin Lee, Ki-Tae Park, and Ju-Hyoung Kim for their help with sampling and analysis.

**Conflicts of Interest:** The authors declare no conflict of interest. The funders had no role in the design of the study; in the collection, analyses, or interpretation of data; in the writing of the manuscript; or in the decision to publish the results.

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
