# Peer review of "The Effects of Ocean Acidification and Warming on Growth of a Natural Community of Coastal Phytoplankton"

_jmse, doi:10.3390/jmse8100821_

Round 1

Reviewer 1 Report

The revision and argument are overall acceptable. The paper can be accepted at the present form.

Reviewer 2 Report

No further comments

This manuscript is a resubmission of an earlier submission. The following is a list of the peer review reports and author responses from that submission.

Round 1

Reviewer 1 Report

This study investigated the combined effects of ocean acidification and warming on the growth of the coastal phytoplankton assemblage in the Repubic of Korea.  This is an interesting work and the manuscript is well written with a clear logical flow to follow.  However, there is no hypothesis in the introduction section, and this section could be better if novelty and knowledge gaps can be further highlighted.  After all, there are already similar studies relating to the effects of ocean warming and acidification on phytoplankton assemblage.  This work may be accepted for publication in the journal after major revisions with the specific comments shown as below.

  1. Line 22. Please check if 400 should be changed to 900 to be consistent with the following main text.
  2. Figure 1. I’m wondering why the authors did not conduct an experiment setup with 400 μatm CO2 + 3 oC warmer than ambient temperature. Such that, the effects of temperature on the phytoplankton assemblage change can be observed.
  3. Line 97. Was pH determined or not?
  4. Figure 2. Is there no SD for control in Fig. 2a and 2b? Is there no data for acidification treatment in Fig. 2b? Why did the authors use three lines for greenhouse treatment in Fig. 2b rather than mean and SD like others?
  5. Figure 3. How to judge if there are significant differences for these parameters among treatments?
  6. Line 192. How to define the dominant species? 0.2% is too low.
  7. Line 227. The dynamic abundance changes of DT and HDF are described in Figures 5 and 6. However, the two classes of phytoplankton only account for 10% of the total. Why did not the authors show the dynamic change of abundance for PP and ANF that account for 90% of the total?
  8. Line 333. In the title, the “assemblage” is used but in the other sections “community” and “communities” are used. Please change and keep consistent.

Reviewer 2 Report

The manuscript describes a well-designed and well-performed mesocosm experiment to study the effects of acidification and temperature increase on the development of a phytoplankton community.

The authors have given a clear description of the results, both in the text and in the figures.

I have no major comments on the paper.

There is only one point that might be useful to consider in the discussion. The experiment ran for approximately 3 weeks, and the results therefore reflect the short-term response of the plankton community. The authors acknowledge that mesozooplankton was excluded from the study because of the potential effects of grazing on the growth of the phytoplankton community (line 301-303). While I agree that mesozooplankton grazing would have been a confounding factor, and the duration of the experiment would probably have been too short anyway to allow a response of mesozooplankton, it makes you wonder how a linger duration of the experiment and inclusion of mesozooplankton grazing would have affected the current findings.